# The Current Treatment Landscape of Malignant Pleural Mesothelioma and Future Directions

**DOI:** 10.3390/cancers15245808

**Published:** 2023-12-12

**Authors:** Beatriz Bertin, Miguel Zugman, Gustavo Schvartsman

**Affiliations:** 1Faculdade Israelita de Ciências da Saúde Albert Einstein, Hospital Israelita Albert Einstein, São Paulo 05651-901, Brazil; beatrizmariasb@gmail.com; 2Department of Medical Oncology, Hospital Israelita Albert Einstein, São Paulo 05651-901, Brazil; zugman.miguel@gmail.com

**Keywords:** mesothelioma, pleura, asbestos, chemotherapy, immunotherapy, targeted agents, cellular therapy

## Abstract

**Simple Summary:**

Malignant pleural mesothelioma is an invasive and drug-resistant tumor related to asbestos exposure, with limited therapy options. It is associated with an unfavorable prognosis and a 5-year survival rate of only 12%. Current standard-of-care treatment based on platinum-pemetrexed chemotherapy has been in place for the past two decades, though survival is increased by just a few months. In this article, we aim to review the current chemotherapy and immunotherapy options for this malignancy and highlight recent developments with regard to chemoimmunotherapy, targeted agents and cellular therapy.

**Abstract:**

The incidence of malignant pleural mesothelioma is expected to increase globally. New treatment options for this malignancy are eagerly awaited to improve the survival and quality of life of patients. The present article highlights the results of recent advances in this field, analyzing data from several relevant trials. The heterogeneous tumor microenvironment and biology, together with the low mutational burden, pose a challenge for treating such tumors. So far, no single biomarker has been soundly correlated with targeted therapy development; thus, combination strategies are often required to improve outcomes. Locally applied vaccines, the expansion of genetically engineered immune cell populations such as T cells, the blockage of immune checkpoints that inhibit anti-tumorigenic responses and chemoimmunotherapy are among the most promising options expected to change the mesothelioma treatment landscape.

## 1. Introduction

Malignant pleural mesothelioma (MPM) is a rare malignancy of the pleural lining highly associated with asbestos exposure (over 80% of the cases) [1]. In spite of its well-established relation with this mineral fiber, the risk of developing the malignancy is only 5% in high-risk populations [2]. The prevalence is higher in men due to occupational exposure, and the median age of diagnosis is 71 years [1]. This aggressive type of tumor undergoes a long latency period (10–40 years), is mostly diagnosed at late stages and has a poor prognosis and a median survival time of one year [3].

The implementation of rigorous regulations in developed countries during the 1990s led to a decrease of over 75% in asbestos usage across various industries. This regulatory approach stands as the foremost strategy for reducing the incidence of MPM within these developed nations. On the other hand, major developing economies have embraced less stringent regulations concerning asbestos, allowing for its ongoing widespread use. This disparity raises concerns about the potential for a global increase in MPM cases [4]. Even within developed nations, the potential for a sustained or increased incidence of MPM is present, primarily stemming from a demographic shift towards an aging population. While asbestos exposure continues to be the predominant risk factor, there are a few other etiologies that may be associated with mesothelioma: exposure to other mineral fibers (erionite, fluoroedenite and balangeroite), radiation, chronic pleural inflammation and germline mutations [5].

A better understanding of MPM’s molecular biology may provide a benchmark for the development of more efficient treatments. The best outcomes so far for patients with mesothelioma have been reported in those who have received multimodal therapy, which usually includes a combination of chemotherapy, surgery and radiation treatment. However, few patients are candidates for this type of approach, and systemic therapy alone is what is feasible to most. Expanding our knowledge with the use of next-generation sequencing and computational technology may further elucidate important pathologic and genomic aspects of the disease, which add to the histologic subtype, and other biomarkers are fundamental to personalizing and determining the optimal treatment [2,6].

MPM’s increasing incidence, aligned with the population aging, high lethality and only modest treatment advances in the past decade, prove to be an unmet medical need. This review highlights the recent advances in the treatment of this aggressive tumor, such as chemoimmunotherapy, and brings to light other promising strategies with the use of targeted agents and cellular therapies.

## 2. Molecular Biology, Genomics and Immunology

MPM can be histologically classified as an epithelioid, which accounts for the majority of cases (around 50%) and has a better prognosis when compared to the sarcomatoid (10% of cases), a more rare, invasive and resistant tumor subtype. The remaining 40% correspond to the biphasic subtype, a mosaic of the previous two [7,8]. However, intra-tumor biology is heterogeneous, and a thorough description of the pathologic specimen may have important prognostic implications. A recent multi-omic study, integrating epigenetic and transcriptomic data, proposed a new method of classification that takes into account not only the histological aspects of MPM but also its microenvironment and inter- and intra-tumor variability. The resulting classification ranks the tumor in a scale from epithelioid to sarcomatoid, creating a continuum of these two populations [9]. A further understanding of tumor biology may also provide novel predictive biomarkers to better inform therapeutic options and clinical trial design, especially in the field of targeted therapies and immunotherapies [9].

Genomic alterations in MPM are primarily related to a loss of function of tumor suppressor genes. BAP1 is the most frequently reported, and others include NF2, CDKN2A, TP53, LATS2 and SETD2 [6,10]. Mesothelioma has a lower tumor mutational burden than most solid tumors, and hence, other predictive immunotherapy biomarkers are warranted [6].

The tumor’s microenvironment, constituting endothelial, stroma and immune cells, has drawn attention as a possible driver of disease, influencing tumor progression, and, therefore, has been under extensive scrutiny for possible therapeutic targets [11]. Some of the immune infiltrating tumor cells are known to be anti-tumorigenic, while others favor tumor growth by dampening the immune response [12]. As an example, lymphocytes (cytotoxic and T helper cells), dendritic cells and natural killer cells are anti-tumorigenic, while myeloid-derived suppressor cells and regulatory T cells are pro-tumorigenic; macrophages and neutrophils have a variable role and can be related to both pro- and anti-tumor activity [13,14]. The mesothelioma microenvironment is subject to the influence of asbestos fiber exposure, which has been linked to the development of an immunosuppressive profile [11]. New therapies should employ a combination of strategies, including immunotherapies that could both inhibit and stimulate specific immune cells of the MPM microenvironment [15].

Pre-clinical models have increasingly played an important role for better understanding MPM’s development and for the investigation of newer interventions and drug testing. However, cell lines, either 2D or 3D spheroids, suffer from similar limitations in which the replication of the true tumor microenvironment and tumor heterogeneity are hard to accomplish. Animal models are helpful in understanding tumorigenesis with specific gene knockouts or asbestos-infused murine pleura, but limitations include the preferential development of aggressive sarcomatoid models in the former and one more similar to human histology in the latter, although unviable for drug testing because of its long latency period [16]. The most promising innovation for the acceleration of drug development in cancer precision medicine may rely on organoids, which have been shown to be applicable in the prediction of cisplatin sensitivity in mesothelioma models [17].

## 3. Current Treatments

The current treatment landscape for MPM emphasizes its palliative intent. The 5-year overall survival estimates are 5–12% at best [18]. Supportive care strategies must encompass pleural effusion and pain management. Surgery is indicated for a small fraction of patients with mesothelioma due to its complexity and high morbidity rate, even in those with a favorable performance status and tumor characteristics [19]. Prior to surgery, pleuroscopy may be needed to elucidate the necessity and feasibility of surgery. Even for patients who are candidates for surgery, the randomized phase 3 trial MARS2 did not show an advantage for patients who were operated on versus those who underwent chemotherapy alone, though some patients can still be individually benefited [20]. Most patients are candidates for systemic treatment, which can improve the survival and quality of life. Palliative radiation can be judiciously used at the physician’s discretion.

### 3.1. Chemotherapy and the Vascular Endothelial Growth Factor Receptor (VEGFR) Pathway

The established treatment for MPM is platinum-antifolate chemotherapy, with a combination of pemetrexed and cisplatin, or carboplatin for patients who cannot tolerate cisplatin. This treatment strategy was established in 2003 in the EMPHACIS trial, which showed an increased median overall survival by 2–3 months when compared to that of cisplatin alone [21,22,23]. Since then, a long gap of treatment approvals was initiated until more recent advances with immunotherapy. An exception occurred with the possible addition of bevacizumab, an anti-VEGF monoclonal antibody, to the treatment regimen. The Mesothelioma Avastin plus Pemetrexed-Cisplatin Study (MAPS) tested a combination of bevacizumab to the present standard of care (pemetrexed and cisplatin) compared to chemotherapy alone, showing an improved median overall survival by 2.8 months and a possible benefit in pain control [24]. Despite the improvement in the quality of life and survival, this treatment regimen has not been filed for an FDA-license [7]. As a second-line regimen, several agents have been tested, with limited activity [25,26,27]. Notably, gemcitabine combined with ramucirumab, an anti-VEGFR-2 monoclonal antibody, improved overall survival compared to gemcitabine alone in a randomized phase 2 trial (RAMES), emphasizing the role of VEGF pathway blockage [28].

### 3.2. Immunotherapy

Several immune checkpoint inhibitors (ICI) are under investigation as potential treatments for mesothelioma. Nivolumab, a programmed cell death 1 (PD-1) inhibitor, was evaluated in the CONFIRM phase III trial, in which patients with refractory disease following platinum-based doublet chemotherapy were given nivolumab or placebo [29]. Patients in the experimental arm showed longer progression-free survival and overall survival with the use of nivolumab. The benefit seemed to be driven by those with tumor-expressing PD-L1, which was more common in non-epithelioid tumors.

Another strategy is to target two immune checkpoints with the association of an anti- PD-1 antibody and anti-cytotoxic T-lymphocyte protein 4 (CTLA-4) antibody. MAPS2 was a non-comparative, randomized phase 2 trial testing the efficacy of nivolumab alone and nivolumab-ipilimumab (anti CTLA-4 antibody) regimens in patients with relapsed disease. The trial showed numerically similar disease control rates, with 44% with nivolumab and 50% with nivolumab-ipilimumab progression-free at 12 weeks [30]. The phase III Checkmate 743 study randomized 600 patients to either cisplatin-pemetrexed or nivolumab-ipilimumab as a first-line treatment [31]. The nivolumab plus ipilimumab regimen had a median OS of 18.1 months compared to 14.1 months of the chemotherapy regimen (HR: 0.73; 95% CI: 0.61–0.87). At three years, the progression-free survival rates were 14% versus 1%. The duration of response at the three-year mark also favored nivolumab plus ipilimumab versus nivolumab alone, 28% versus 0%, respectively. At four years, the OS rates were 17% versus 11% [32]. The results indicated that nivolumab plus ipilimumab should be the standard-of-care treatment for unresectable MPM based on the evidence of a longer survival benefit over chemotherapy regardless of tumor histology, though the benefit was particularly more pronounced in the sarcomatoid subtype. The outcomes of this trial for unresectable MPM led the FDA to approve the use of nivolumab/ipilimumab as a first-line treatment.

Another potential treatment uses pembrolizumab, an anti-PD-1 antibody [33], which has been tested in the KEYNOTE-028 phase I trial in patients with MPM. As an early phase trial, it presented promising results; however, the PROMISE-meso phase 3 trial was negative in the second-line setting [33,34].

Table 1 summarizes the trials with relevant results related to the current practice.

## 4. Novel Treatments for MPM

### 4.1. Chemoimmunotherapy

A phase II, multicenter, single-arm trial (DREAM), conducted with 54 previously untreated patients, evaluated the combination of durvalumab (anti-PD-L1 antibody) and chemotherapy with cisplatin and pemetrexed for advanced MPM [35]. The results were promising, with 57% of patients alive and progression-free at 6 months.

The IND227 phase 3 trial tested the combination of pembrolizumab with pemetrexed-platinum chemotherapy to chemotherapy alone, showing a statistically significant OS benefit with the combination (HR: 0.79; 95% CI: 0.64–0.98; *p* = 0.0324), with a median OS of 17.3 months compared to 16.1 months in the control arm. The three-year survival rates were also higher in the experimental arm (25% against 17%). The study showed that the addition of pembrolizumab to platinum-pemetrexed improves the overall response rate from 38% to 62%, with no new safety concerns. Exploratory analysis indicates again that non-epithelioid histologies may benefit the most from the addition of immunotherapy to the treatment regimen [36,37].

The Bevacizumab and Atezolizumab in Malignant Pleural Mesothelioma (BEAT-meso), a randomized phase III trial, is assessing the efficacy of atezolizumab (a PD-L1 blocker) combined with bevacizumab in addition to standard chemotherapy compared to the administration of bevacizumab and chemotherapy [38]. The study has recruited 400 patients across Europe, and its results are expected to be released in 2024. Similarly, the DREAM3R, a phase III randomized trial, is evaluating the use of anti PD-L1 and durvalumab, in combination with cisplatin and pemetrexed, for the first-line treatment of advanced MPM [39].

### 4.2. Novel Immunotherapy Approaches

#### 4.2.1. Immune Checkpoints

New emerging immune checkpoints, such as lymphocyte activation gene-3 (LAG-3), are being evaluated in MPM. LAG-3 is expressed on the surface of T cells, whose negative regulatory role hampers T-cell activation and proliferation against tumor antigens [40]. It has been shown that LAG-3 is expressed on immune cell infiltrates isolated from patients with MPM. Pre-clinical models have shown delayed tumor growth and a survival benefit in mice with the administration of an anti-PD-1 plus anti-LAG-3 antibody [41,42]. A phase I trial designed to assess the safety and tolerability of tebotelimab, a bispecific antibody designed to bind PD-1 and LAG-3 and restore the function of exhausted T cells in advanced solid tumors, showed encouraging preliminary results [43]. VISTA is another relevant immune checkpoint, expressed mostly by epithelioid MPM tumors, and is being investigated as a potential target for MPM treatment in several studies that combine the anti-VISTA antibody with vaccines and other ICIs [44,45].

#### 4.2.2. Oncoviral Therapy

Oncoviral therapy also represents a potential line of treatment for mesothelioma. Modified viruses, such as adenovirus or measles virus containing human genes, are injected into patients to induce polyclonal anti-tumor activity by their own immune system [46]. A phase II trial with 40 patients demonstrated the safe and feasible results of administering an intrapleural injection of a non-replicating adenoviral vector (Ad) expressing the immune-activating cytokine interferon-alpha (IFN) in patients with MPM, followed by celecoxib and chemotherapy. Celecoxib is an inhibitor of the immunosuppressive molecule PGE2 used to further manipulate the tumor microenvironment. The regimen causes a large production of interferon in the pleura, translated into an intense stimulus to the patient’s immune system, with a promising disease control rate of 88% [46]. A larger, randomized phase III trial (INFINITE) is currently underway, testing the administration of adenovirus-delivered Interferon Alpha-2b (rAd-IFN) in combination with celecoxib and gemcitabine in 53 patients with MPM [47]. The results are expected by the end of 2024.

#### 4.2.3. Cellular Therapy

##### CAR-T

Cellular therapy involving Chimeric Antigen Receptor (CAR) T cells has been proven to be a successful treatment for hematological tumors but still has not presented compelling evidence in the treatment of solid tumors due to its heterogeneous nature [48]. For T-cells to be able to exert anti-tumoral activity, they need to fulfill several steps. They must infiltrate the tumor tissue and be activated against tumor antigens. T-cell therapies are challenged with some of the following barriers: (i) an immunosuppressive microenvironment of solid tumors imposes resistance to T-cell therapy, (ii) an expression of PD-L1 in tumor cells inactivates T-cells and (iii) genomic instability leads to tumor cell heterogeneity, with different clonal populations expressing different antigens [49]. To overcome such barriers, researchers have: (i) applied CAR-T cells regionally on the pleura to increase tumor infiltration, a strategy that granted better success rates compared to intravenous infusion, (ii) carried out the blockage of inhibitory signals by tumor cells and the tumor microenvironment and (iii) focused on using targets for CAR-T cell therapy that are less expressed in healthy tissues but overexpressed in MPM cells, which is the case for the antigen mesothelin (MSLN)—overexpressed in 80–90% of MPM [49]. So far, phase I/II clinical trials using anti-MSLN CAR T cells to treat MPM have been promising, showing anti-tumor activity and good safety outcomes.

CAR-T cell therapy is likely the most promising treatment strategy compared to other targeted therapies. Once infused, CAR-T cells multiply and persist in the patient’s body, which may overcome the tumor’s immune tolerance and promote long-term immune surveillance, preventing recurrence through immune-reactivation once re-encountering the tumor’s antigens [50]. CAR-T cell therapy may be enhanced with the combination of ICI therapies, such as PD-1 or PD-L1 blockade, further preventing tumors from immune evasion. A phase I trial demonstrated that the intrapleural administration of MSLN CAR T-cell followed by a PD-1 blockade (pembrolizumab) in pretreated patients with MPM was feasible and well tolerated [51]. Furthermore, patients who received the combined treatment had a median overall survival of almost two years compared to 17.7 months in patients who received only CAR-T cells. Several groups are currently conducting trials evaluating different CAR-T cell products in mesothelin-expressing tumors [52,53,54].

##### Dendritic Cell Therapy

Although checkpoints inhibitors have been shown to improve outcomes for MPM patients, only a few derive significant benefits from immunotherapy. The use of PD-1 or PD-L1 inhibitors is expected to activate the T-cell killing capacity [55]. In this sense, a low density of CD8+ T-cells may limit its single-agent activity [56]. CD8+ T-cell infiltration positively correlates with a better overall survival in MPM patients.

Dendritic cell (DC) therapy aims to induce the proliferation of T cells and promote the activation of CD4+ and CD8+ T-cells by presenting them with tumor antigens, allowing CD8+ T-cells to infiltrate the tumor microenvironment [57]. DC can be derived from the patient’s bone marrow or peripheral blood through an ex vivo maturation process stimulated by cytokines. Matured DCs are loaded with tumor antigens (peptides, lysate and others) that are processed by the cell and transferred to its surface (through MHC I and II molecules). These processed DC cells are then transferred back to the patient to stimulate the immune response against the tumor [58].

Phase I trials have demonstrated that autologous tumor lysate-pulsed DC immunotherapy increased the T-cell response against MPM; however, using autologous tumor material imposes many challenges to conducting larger clinical trials [59]. Therefore, efforts have been made to verify the plausibility of using allogeneic tumor lysate as an antigen source. The DENIM randomized phase II/III trial is assessing the efficacy of autologous DCs loaded with allogeneic tumor lysate as a potential maintenance treatment for MPM following first-line treatment with chemotherapy in patients who had not shown disease progression [57].

### 4.3. Targeted Agents

#### 4.3.1. EZH2

The hyperexpression of the enhancer of zeste homolog 2 (EZH2) is related to cancer progression. EZH2 is a subunit of the oncogenic polycomb repressive complex 2, frequently present in association with BAP1 loss [36]. Mesothelioma cells with inactivated BAP1 are sensitive to EZH2 pharmacologic inhibition, a fact that led investigators to launch a phase II trial to assess the effect of tazemetostat, an EZH2 inhibitor, in relapsed MPM patients with inactivated BAP1 [60,61]. The disease control rate was 54% at week 12 (primary outcome of the trial) and 28% at 24 weeks. Tazemetostat also presented a favorable toxicity profile [61].

#### 4.3.2. ASS1

Arginine is an amino acid synthesized by cells and is essential for their growth. Notwithstanding, some tumors lack an important enzyme in the process of synthesizing arginine, called argininosuccinate synthetase 1 (ASS1), depending on the exogenous supply [62]. Lower ASS1 expression has been associated with more aggressive tumors and worse prognoses in different malignancy types, including mesotheliomas [62]. Arginine deprivation, therefore, is currently being evaluated by several studies as a potential therapy. Arginine-depleting agent (ADI-PEG 20) has already presented promising outcomes in treating patients with MPM in a phase I trial combined with cisplatin and pemetrexed chemotherapy, paving the way for a randomized phase II/III trial to scale this potential therapy (ATOMIC-Meso Phase 2/3 Study) [63]. ATOMIC recruited 249 MPM patients and is currently investigating the safety and efficacy of the same treatment regimen tested in the phase I trial mentioned before.

#### 4.3.3. Molecular-Stratified Therapy

The Mesothelioma Stratified Therapy (MiST) is a multicenter ongoing clinical trial being conducted in the United Kingdom (UK), trying to identify predictive biomarkers and evaluate new personalized therapy for mesothelioma [64]. It seeks to stratify patients based on the molecular characteristics of their disease to better individualize treatment strategies. MiST has been designed with three different arms.

MiST1: Patients in this arm were selected based on mutations in the BRCA-1 or BAP1, known to be found in MPM tumors. BAP1, similar to the BRCA1 gene, is involved in DNA repair and can potentially be targeted with the use of poly-ADP ribose polymerase inhibitors (PARPi). The Mesothelioma Stratified Therapy 1 (MiST1) is a phase II trial studying the use of rucaparib, a PARPi, in 26 patients diagnosed with relapsed mesothelioma with BAP1 or BRCA-1 deficiency. The results showed some activity, with manageable toxicity, with a 58% disease control rate at 12 weeks and one of 23% at 24 weeks [65].

MiST2: MiST2 is a phase II trial focused on p16ink4A-negative mesothelioma previously treated with chemotherapy. The loss of the gene CDKN2A, frequently found in mesotheliomas, is associated with poorer prognosis due to the loss of the tumor suppressor p16ink4A, an endogenous suppressor of cyclin-dependent kinase (CDK)4 and CDK6. The trial is investigating the use of abemaciclib, an inhibitor of CDK4/6, in 26 patients; the results have also shown a 54% disease control rate at 12 weeks [66].

MiST3: The third arm trial, MiST3, will test the inhibition of AXL, a member of the TAM (Tyro3, AXL, Mer) family of receptor tyrosine kinases. AXL is a key regulator of tumor plasticity and immune evasion, contributing to tumor-intrinsic and microenvironmental immune suppression [67]. The overexpression of AXL in 74% of mesothelioma tumors examined by an analysis led to an ongoing trial investigating the potential of bemcentinib, an AXL inhibitor, combined with pembrolizumab in patients with relapsed mesothelioma.

### 4.4. Tumor-Treating Fields

In 2019, The Food and Drug Administration (FDA) approved the NovoTTF system, a humanitarian use device, to be used in combination with first-line standard chemotherapy (platinum-pemetrexed) for the treatment of MPM. NovoTT is a device based on the application of specific electric frequencies (tumor treatment fields, TTF) to diminish cancer growth [6,68]. This approval occurred sooner than expected, since the results from a randomized phase III trial have not yet confirmed the results of the phase II STELLAR trial, a single-arm study with 80 patients conducted in Europe, which demonstrated that TTF combined with chemotherapy had an overall survival of 18.2 months [69].

Table 2 summarizes the most relevant trials related to novel treatment pathways.

## 5. Discussion

The global incidence of MPM has been suffering upward pressure due to the widespread use of asbestos by industries in large developing economies and the aging population shift in developed countries. The aggressiveness of the disease also drives researchers to look for more favorable treatments that could improve survival and reduce morbidity [4]. Given the proven efficacy of chemotherapy and immunotherapy, the possibility of achieving better outcomes from combining both strategies led researchers to investigate novel therapies. However, the low mutational burden coupled with a diverse tumor microenvironment and biology pose challenges to defining a predictive biomarker for more suitable therapeutic options [9].

Mesothelioma has an immunosuppressive profile; hence, focus may be on boosting immune cells to increase the anti-tumorigenic response and on inhibiting pro-tumorigenic cell functions. In this direction, several immunotherapy strategies are being evaluated, but still with small practical breakthroughs (Figure 1). Nivolumab/ipilimumab is the only one approved by the FDA as a first-line treatment for unresectable MPM. The FDA’s decision was supported by the results of the Checkmate 743 trial, which showed a 4-month increase in overall survival compared to chemotherapy regardless of the tumor histology type [31].

Clinical trials have been assessing the potential of chemoimmunotherapy, combining standard-of-care chemotherapy (platinum-pemetrexed) with different immunotherapeutic strategies. The addition of pembrolizumab (anti-PD-1 antibody) to chemotherapy seems to improve overall survival by only a month, with a more pronounced effect in non-epithelioid histologies, as reported by the IND227 trial [37]. Several ongoing phase III trials may reveal interesting options of combining available ICI and anti-VEGF agents with chemotherapy.

The evolution of immune oncology has led to the discovery of several other checkpoints currently being evaluated in most solid tumors, including MPM. LAG-3 and VISTA are among promising proteins to be targeted. The use of vaccines to stimulate the immune system coupled with chemotherapy is another avenue that could lead to positive outcomes. Delivering such therapies locally in the pleura may enhance their potential while minimizing systemic toxicities [15]. Even though preliminary data on oncoviral therapy regionally applied are favorable, more sound results are still necessary. The phase III trial INFINITE is testing adenovirus-delivered Interferon Alpha-2b efficacy and should indicate whether oncoviral treatment may be used as MPM therapy [47]. Similarly, CAR-T cells directed to mesothelin, locally administered, are expected to improve outcomes. Dendritic cell therapies may also increase CD8+ T-cell density and enhance anti-PD-1 activity.

Even though the search for biomarkers has been under a lot of focus, there does not seem to be a single driver alteration that is amenable to targeted therapy. The loss of tumor suppressor genes, mainly BAP1 and CDKN2A, is the most predominant genomic alteration in MPM, and it could represent an important biomarker. In this sense, agents targeting PARP enzymes, CDK4/6 and AXL are being evaluated.

## 6. Conclusions

The recent advances in understanding the immune landscape and molecular profile of MPM allowed for several agents used in different scenarios to be investigated for this disease. Moreover, novel biomarker-directed therapies are being developed to target specific mechanisms of mesotheliomas. The complexity and heterogeneity of this deadly disease may have an increased chance of success with combined approaches.

## Figures and Tables

**Figure 1 cancers-15-05808-f001:**
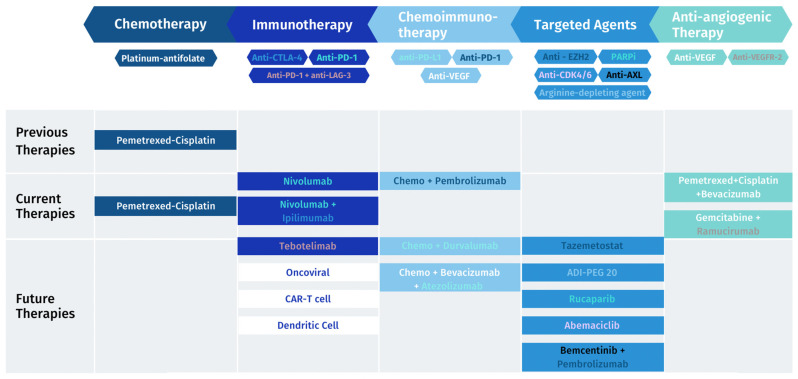
MPM treatment landscape: previous, current and future therapies.

**Table 1 cancers-15-05808-t001:** Relevant clinical trials on current MPM treatment.

Study Name	Description	Treatment	Study Completion	ClinicalTrials.gov Number
MARS2	Multicenter open parallel group randomized controlled trial comparing the effectiveness of surgery—(extended) pleurectomy decortication—versus no surgery for the treatment of pleural mesothelioma	Surgery + chemotherapy vs. chemotherapy	2022	NCT02040272
EMPHACIS	Phase III trial determining whether treatment with pemetrexed and cisplatin results in a survival time superior to that achieved with cisplatin alone	Pemetrexed + cisplatin vs. cisplatin alone	2003	-
MAPS	Randomized, controlled, open-label, phase III trial assessing the effect on survival of bevacizumab when added to the present standard of care, cisplatin plus pemetrexed, as a first-line treatment of advanced MPM	Bevacizumab + pemetrexed + cisplatin vs. pemetrexed + cisplatin alone	2016	NCT00651456
RAMES	Randomized, double-blind, placebo-controlled, phase II trial assessing the efficacy and safety of the anti-VEGFR-2 antibody ramucirumab combined with gemcitabine in patients with pretreated MPM	Gemcitabine + ramucirumab vs. gemcitabine + placebo	2020	NCT03560973
CONFIRM	Multicenter, placebo-controlled, double-blind, parallel group, randomized, phase III trial assessing the efficacy and safety of nivolumab, an anti-PD-1 antibody, in patients with pleural or peritoneal malignant mesothelioma who have progressed following platinum-based chemotherapy	Nivolumab vs. placebo	2023	NCT03063450
MAPS2	Multicenter randomized, non-comparative, open-label, phase II trial prospectively assessing the anti-PD-1 monoclonal antibody alone or in combination with the anti-cytotoxic T-lymphocyte protein 4 (CTLA-4) antibody in patients with MPM	Nivolumab + ipilimumab vs. nivolumab alone	2019	NCT0271627
CHECKMATE 743	Open-label, randomized, phase III study testing the effectiveness and tolerability of the combination of nivolumab and ipilimumab compared to pemetrexed and cisplatin or carboplatin in patients with unresectable pleural mesothelioma	Nivolumab + ipilimumab vs. pemetrexed + cisplatin/carboplatin	2023	NCT0289929

MPM: malignant pleural mesothelioma.

**Table 2 cancers-15-05808-t002:** Relevant clinical trials on novel MPM treatment.

Study Name	Description	Treatment	Study Completion	ClinicalTrials.gov Number
**Chemoimmunotherapy**				
DREAM	Multicenter, single-arm, open-label, phase 2 trial evaluating the activity of durvalumab, an anti-PD-L1 antibody, given during and after first-line chemotherapy with cisplatin and pemetrexed in patients with advanced MPM	Durvalumab + pemetrexed + cisplatin	2019	ACTRN12616001170415 *
IND227	Phase 2 trial comparing the progression-free survival of standard platinum and pemetrexed (CP) versus CP + pembrolizumab	Platinum + pemetrexed vs. platinum + pemetrexed + pembrolizumab	2023	NCT02784171
BEAT-meso	Multicenter randomized phase III trial comparing atezolizumab plus bevacizumab and standard chemotherapy versus bevacizumab and standard chemotherapy as first-line treatments for advanced MPM	Bevacizumab + pemetrexed + carboplatin vs. bevacizumab + pemetrexed + carboplatin + atezolizumab	2024	NCT03762018
DREAM3R	Phase III randomized trial aiming to determine the effectiveness of including durvalumab with first-line platinum-pemetrexed chemotherapy in advanced MPM	Durvalumab + pemetrexed + cisplatin/ carboplatin vs. pemetrexed + cisplatin/ carboplatin alone	2025	NCT04334759
**Novel Immunotherapies**				
INFINITE	A Phase 3, open-label, randomized, parallel group study evaluating the efficacy and safety of the intrapleural administration of adenovirus-delivered interferon Alpha-2b (rAd-IFN) in combination with celecoxib and gemcitabine in patients with MPM	rAd-IFN + celecoxib + gemcitabine vs. celecoxib + gemcitabine alone	2024	NCT03710876
A Phase I/II Clinical Trial of MPD Treated With Autologous T Cells Genetically Engineered to Target the Cancer-Cell Surface Antigen Mesothelin	Open-label, dose-escalating, non-randomized, single-center, phase I/II study of mesothelin-targeted T cells administered intrapleurally as an infusion in patients with a diagnosis of MPD from mesothelioma, lung cancer or breast cancer	CAR T-cell + pembrolizumab	2024	NCT04577326
DENIM	Open-label, multicenter, randomized phase II/III trial patients will be randomized to receive either dendritic cell therapy plus best supportive care (BSC) or BSC alone according to the discretion of the local investigator after first-line chemotherapy treatment.	Dendritic cell therapy + BSC vs. BSC alone	2023	NCT03610360
**Targeted Agents**				
A Multicenter Study of the EZH2 Inhibitor Tazemetostat in Adult Subjects With Relapsed or Refractory Malignant Mesothelioma With BAP1 Loss of Function	Phase 2, multicenter, open-label, two-part, single-arm, two-stage study aiming to evaluate the anti-tumor activity and safety of tazemetostat in patients with measurable relapsed or refractory MPM	Tazemetostat	2019	NCT02860286
ATOMIC-Meso	Randomized, double-blind, phase II/III study in subjects with MPM assessing the efficacy of ADI-PEG 20 combined with pemetrexed and cisplatin	ADI-PEG20 + pemetrexed + cisplatin vs. placebo + pemetrexed + cisplatin	2022	NCT02709512
Mesothelioma Stratified Therapy (MiST)	Stratified multi-arm phase IIa clinical trial enabling the accelerated evaluation of targeted therapies for relapsed malignant mesothelioma. Stage 1: molecular pre-screening for the identification of patients, biomarker testing and analysis. Stage 2: the treatment protocol will be specific to the patient based on the results of their biomarker testing in stage 1 **. Stage 3: molecular profiling to understand the genomic basis of the drug response in the MiST trial	Rucaparib ademaciclib pebrolizumab + bemcentinib Atezolizumab + bevacizumab dostarlimab + niraparib	2023	NCT03654833
**Tumor-treating Fields**				
STELLAR	Prospective, single-arm, non-randomized, open-label phase II trial designed to study the safety and efficacy of a medical device, the NovoTTF-100L, concomitant with pemetrexed and cisplatin or carboplatin in MPM patients	TTFields at a frequency of 150 kHz to the thorax + pemetrexed + platinum/ carboplatin	2018	NCT02397928

* This study is registered with the Australia New Zealand Clinical Trials Registry. ** Treatment options are described on the treatment column. Drug names of all possible treatments in the trial. MPM: malignant pleural mesothelioma; TTFields: tumor treatment fields; MPD: malignant pleural disease.

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
