# Peer review of "The Current Treatment Landscape of Malignant Pleural Mesothelioma and Future Directions"

_cancers, 2023, doi:10.3390/cancers15245808_

Round 1

Reviewer 1 Report

Comments and Suggestions for Authors

              The patients of malignant pleural mesothelioma (MPM) are increasing worldwide. Due to therapy-resistance of MPM, new therapeutic strategy is hoped. In this manuscript, Bertin B et al summarize the current treatment of MPM. This manuscript is well-organized; however, following points should be clarified.

Major points

#1: Authors emphasize the immunotherapy. Pleurectomy may be needed to refer for current surgery.

#2: In page 2 line 79-80, Please show the citation of the warrant of predictive immunotherapy biomarker and evidence of a low tumor mutational burden. What cancer type did authors compared to MPM?

Minor points

##1: The figure of milestone may help readers to understand the previous, current and future therapy for MPM.

Comments on the Quality of English Language

English is fine

Reviewer 2 Report

Comments and Suggestions for Authors

The ms reviews the current standard of care for malignant mesothelioma.

I just suggest to explore new therapy directions, also at in vitro or at preclinical stage.

Comments on the Quality of English Language

just few english errors to avoid

Reviewer 3 Report

Comments and Suggestions for Authors

Even though malignant pleural mesothelioma is characterized as rare malignant tumor, however, based upon the epidemic feature of this disease, i.e. its diagnosis might be delayed by more than 30 up to 50 years after the first exposure to asbestos, and the incidence is being increased in both developing and developed countries even if the industry use of asbestos has been banned worldwide. So, notably, this review would have a great clinical impact and significance in the field and guidance for bench and bedside researchers/investigators.

A early diagnosis and distinguish from inflammation as well as lung cancer based on the exposure history of asbestos need to be addressed clearly since the prevention of malignant pleural mesothelioma must be the most important than the therapy.

In order to unnecessary misunderstanding and suspicion, it would be appropriate to not mention the particular country's name in the introduction (line 40), instead of developing country is enough;

Also, a few typos (line 156, chemoimnotherapy) need to be corrected before acceptance.

The page# of cited references should be listed according to the journal required (full page#).

Comments on the Quality of English Language

Quality of English is fine but a few typos.
